# Effects of Common Litter Management Practices on the Prevalence of *Campylobacter jejuni* in Broilers

**DOI:** 10.3390/ani12070858

**Published:** 2022-03-29

**Authors:** Matthew A. Bailey, Dianna V. Bourassa, James T. Krehling, Luis Munoz, Kaicie S. Chasteen, Cesar Escobar, Kenneth S. Macklin

**Affiliations:** Department of Poultry Science, Auburn University, Auburn, AL 36849, USA; dvb0006@auburn.edu (D.V.B.); krehljt@auburn.edu (J.T.K.); lrm0029@auburn.edu (L.M.); ksc0030@auburn.edu (K.S.C.); cze0034@auburn.edu (C.E.); macklks@auburn.edu (K.S.M.)

**Keywords:** *Campylobacter*, litter, broiler, compost, sodium bisulfate, cross-contamination

## Abstract

**Simple Summary:**

The bacterium *Campylobacter* is a significant cause of foodborne illness, causing over one million cases per year in the United States. *Campylobacter* is naturally found in chickens and can contaminate chicken products; therefore, strategies to lower *Campylobacter* presence in chickens are important to public health. Commercial chickens are raised in houses with bedding material, or litter, covering the floor. Litter can become contaminated with *Campylobacter*, which in turn will then colonize the birds. In some countries, after a flock of chickens is harvested, the litter is treated and reused for the next flock, which could spread *Campylobacter*. The goal of this study was to observe if reusing contaminated litter could indeed spread *Campylobacter* and to determine if common litter treatments were able to prevent contamination of the next flock. To determine this, previously used litter contaminated with *Campylobacter* was composted and treated with sodium bisulfate. A flock was raised on this litter and tested for *Campylobacter* for 42 days. No *Campylobacter* was detected in any of these samples, indicating that re-used litter is not a probable source for *Campylobacter* contamination of chickens.

**Abstract:**

*Campylobacter* is an important foodborne pathogen and is naturally found in chickens. During broiler production, litter can become contaminated with *Campylobacter* when birds defecate, and this litter, in some countries, is typically reused for the next flock, potentially causing cross-contamination. The goal of this experiment was to observe if reusing contaminated litter could spread *Campylobacter* between flocks and to observe if common litter treatments could prevent this cross-contamination. To determine this, a flock of birds was inoculated with *Campylobacter jejuni* and allowed to naturally contaminate the litter for 42 days. After grow-out, birds were terminated, and litter was given five treatments: uninoculated fresh litter, untreated re-used litter, composted re-used litter, re-used litter treated with sodium bisulfate (45 kg/305 m^2^), and re-used litter composted and treated with sodium bisulfate (45 kg/305 m^2^). A second flock was placed on the litter, grown for 42 days, and tested for *C. jejuni* prevalence. Following inoculation of the first flock, high prevalence of *C. jejuni* was observed; however, after a 19-day down-time between flocks, no *C. jejuni* was detected in any samples from the second flock. These results indicate that re-used litter was not a significant reservoir for cross-contamination of broilers when provided a significant down-time between flocks.

## 1. Introduction

The control of foodborne pathogens and disease is a serious issue to food industries, consumers, and regulators. One of these important diseases is campylobacteriosis, which causes an estimated 1.5 million cases annually in the United States [1] and 96 million cases per year worldwide [2]. *Campylobacter* is associated with poultry products, and the contamination of retail chicken has been reported over a range of 14–89% [3]. In addition, some estimates attribute as many as 30% of campylobacteriosis cases to broiler chicken consumption and up to 80% of cases have been associated with the entire chicken reservoir [4]. These observations suggest that public health would greatly benefit from better control measures in chicken production systems and final products. An important strategy for control is to prevent the contamination of flocks with *Campylobacter* during live production. Strict efforts to maintain biosecurity and hygiene practices on farms are associated with fewer *Campylobacter*-positive flocks [5]. However, these efforts have not been able to eliminate *Campylobacter* from all farms because employees must continuously follow the correct protocols to maintain biosecurity, which is difficult to achieve [6]. In addition, *Campylobacter* is a commensal microorganism in chickens and many other animals and insects, thus it is difficult to remove from a flock after contamination [6]. Once *Campylobacter* has colonized a flock, it spreads rapidly by horizontal transmission [7]. For control to be possible, the common routes of exposure must be identified and interventions developed to mitigate these routes.

One possible route of exposure is used litter, which has been associated with the risk of increased *Campylobacter* prevalence [8], and some researchers have isolated *Campylobacter jejuni* from litter samples collected at commercial farms with a prevalence of 100% [9]. Studies have successfully implemented used litter contaminated with *Campylobacter* from a previous flock to inoculate chicks for testing litter treatments against *Campylobacter* [10], and *Campylobacter* have demonstrated better survival in used litter compared to fresh litter in bioluminescence imaging experiments [11]. These collective studies suggest that used litter is a potential reservoir for the contamination of new flocks with *Campylobacter*. Because commercial broiler farms in some countries typically reuse litter for many flocks, management strategies for mitigating this risk are warranted. Although the re-use of litter has been explored as a vector for *C. jejuni* cross-contamination, there has been little investigation into the effects of commercial litter management practices on the risk of *C. jejuni* cross-contamination when re-using litter. To address this need, the objectives in this study included the following: (1) the generation of used litter with natural levels of *C. jejuni* contamination by inoculating one flock, followed by (2) an assessment of the effects of windrow composting and sodium bisulfate on the prevalence of *C. jejuni* cross-contamination in a second flock raised on the same litter.

## 2. Materials and Methods

### 2.1. Simulating Natural Contamination of Poultry Litter

A pen trial was conducted using fresh pine shavings in 25 floor pens, with standard commercial corn-soybean feeds (starter, grower, finisher) and water provided ad libitum. The starter diet was fed as crumbles and the grower and finisher diets were pelleted. The crude protein and metabolizable energy profiles for feeds are listed in Table 1. Day-old chicks (Ross 308 males) were wing-banded and placed in pens at a density of 50 birds/2.3 m^2^ pen (1250 total birds in each pen, except for 5 negative control pens). Following this, 5 birds were inoculated on day 7 with 0.1 mL of a cocktail containing three *C. jejuni* marker strains resistant to ciprofloxacin (strains designated as 1-3CR32, 4-1CR16, and 5-17CR08) at a concentration of approximately 7.0 log_10_(colony forming units) mL^−1^. The wing band numbers of the inoculated birds were recorded to differentiate them from the un-inoculated birds. This partial-population inoculation strategy, or seeder method, was used to simulate the gradual spread of *C. jejuni* through the flock by horizontal transfer, which is suspected to be the natural route of spread on commercial broiler farms. Seeder methods have been used to successfully inoculate birds with *C. jejuni* in past studies [12,13]. Pens were separated from each other by wire mesh, and the negative control pens were separated from the inoculated pens by an empty pen.

### 2.2. Monitoring Spread of C. jejuni

On days 7, 14, 21, 28, and 42 of the first flock, litter samples were collected from each pen with boot cover swabs (Envirobootie, Hardy Diagnostics, Santa Maria, CA, USA) by walking across the entire surface of each pen. Because the *C. jejuni* prevalence was high after the first few weeks, sampling was not performed on day 35. In addition, ceca samples were collected from 5 un-inoculated birds/pen on the same days to determine the prevalence of birds that were colonized by *C. jejuni* via horizontal transmission. Samples were stored on ice in a cooler for transfer to the lab.

### 2.3. Microbiological Analysis

The prevalence of *C. jejuni* was determined in each sample by first enriching boot swabs and ceca contents in approximately 100 mL of Campylobacter Enrichment Broth (3 M, Saint Paul, MN, USA). Enrichments were incubated for 24 h at 42 °C. After incubation, enrichments were screened for *Campylobacter* using the 3 M Molecular Detection System (MDS) (3 M). Positive enrichments were streaked for isolation onto Campy Cefex agar plates (Hi-Media Laboratories LLC, Mumbai, India) supplemented with 5% horse blood, cycloheximide (200 mg/L), cefoperazone (32 mg/L), and ciprofloxacin (1 mg/L for selection of marker strains). The plates were then incubated for 48 h at 42 °C under microaerobic conditions (5% oxygen, 15% carbon dioxide, and 80% nitrogen). After incubation, plates were examined for typical colonies to confirm that the samples were positive for the *C. jejuni* marker strains.

### 2.4. Litter Treatments

Five treatments were tested to determine the effectiveness of common litter management strategies (Table 2). The treatments consisted of a negative control (treatment A) in which fresh pine shavings were used in pens that had non-inoculated birds for the first flock, a positive control (treatment B) which contained used litter with no treatment, used litter treated with sodium bisulfate (Jones-Hamilton Co., Walbridge, OH, USA) at 45 kg/305 m^2^ between the two flocks (treatment C), used litter treated by windrow composting [14] between flocks (treatment D), and used litter treated with a combination of sodium bisulfate treatment (45 kg/305 m^2^) and windrow composting between flocks (treatment E). Windrow composting is commonly used in the poultry industry to reduce the microbial load in used litter so that pathogens are destroyed, and the litter may be safely re-used for a subsequent flock of chickens [14]. To begin this process, litter is piled up so that the heat generated by microbial degradation of organic materials is trapped in the pile [14]. The temperature must be maintained above 50 °C for at least 1 day to destroy bacteria, and the compost pile is turned periodically to re-introduce oxygen and allow more heat to be generated [14]. In this study, each treatment consisted of 5 replicate pens (25 pens total) with 50 birds/pen (1250 birds total). Windrow composting for treatments C and E was performed for 19 days between flocks 1 and 2. To ensure enough material for proper litter heating, litter from pens that held treatments C and E pens were piled together into a single windrow and turned on day 7 of composting. Portable data logger probes (OM-EL-USB-1-Series, Omega Engineering Inc., Norwalk, CT, USA) were used to monitor the temperature inside the windrow during composting. Sodium bisulfate for treatments D and E was applied to the litter the day before chicks for flock 2 were housed.

### 2.5. Evaluating Effect of Treatments against C. jejuni Prevalence

The second flock also consisted of 50 birds/pen and was raised with the same conditions and feed formulations as the first flock, with the exception of the re-used, treated litter for the treatment groups. Litter from each pen and ceca samples from 5 birds/pen were collected on days 7, 14, 21, 28, and 42 by the same methods described above. In addition, litter samples were collected on day 0 to assess the *C. jejuni* prevalence in the litter before the chicks were housed. Samples were evaluated for the prevalence of *C. jejuni* using the same microbiological methods as described for flock 1.

## 3. Results

### 3.1. Flock 1 C. jejuni Prevalence

The prevalence of *C. jejuni* in the litter samples was 0% on day 7 of the first flock, but was 100% by day 14, or one week after inoculation of the chicks (Figure 1). The litter from the non-inoculated pens also showed a 100% prevalence of *C. jejuni*. The prevalence in litter remained at 100% for the remainder of flock 1 (Figure 1). For ceca samples from flock 1, the prevalence was 0% on day 7 before inoculation (Figure 2). On day 14 of flock 1, or one week after inoculation, the prevalence increased in ceca samples to 81.6% and steadily increased each week to a final prevalence of 94.4% by day 42 (Figure 2). This prevalence included samples from the negative control pens, which showed the same prevalence as the inoculated pens.

### 3.2. Flock 2 C. jejuni Prevalence

For flock 2, zero litter samples tested positive for *C. jejuni* during the entire experiment. For ceca samples, there was a single ceca sample from treatment B (positive control) on day 14 that tested positive by MDS analysis; however, no other samples tested positive for *C. jejuni* during the remainder of the experiment, and no typical colonies were observed on Campy Cefex plates.

## 4. Discussion

The prevalence for flock 1 followed a pattern commonly described in the literature. It has been observed that *C. jejuni* rapidly spreads throughout a poultry house by horizontal transmission once it is introduced [8,15,16]. Our observation of zero prevalence in flock 1 samples prior to inoculation, followed by a high prevalence (100% in litter and >80% in ceca) after inoculation supports this assertion. The fact that the negative control pens were positive although they were not inoculated with *C. jejuni* also illustrates the rapid spread of this organism and shows that separation by pens within a house is not sufficient to prevent cross-contamination. In addition, the horizontal transmission model is also supported by the use of a seed-inoculation method in this experiment, which suggests that *C. jejuni*-positive birds were colonized horizontally by *C. jejuni* shed from inoculated birds. Inoculated birds were not tested for *C. jejuni* prevalence; however, because marker strains were used, it is unlikely that *C. jejuni*-positive birds were colonized by another source.

The prevalence for flock 2 was zero for nearly the entire experiment. Although one sample tested positive via MDS analysis, this occurred late in the analysis (over 30 min), indicating a potential false positive (MDS does not report actual cycle threshold numbers), as an enriched sample should show a positive result much earlier. Given that MDS analysis does not distinguish between live and dead cells [17], and considering the tendency for *C. jejuni* to spread rapidly upon introduction, it appears that this was indeed a false positive. In addition, no typical colonies were observed after streaking the enrichments onto Campy Cefex.

Because no *C. jejuni* was detected in flock 2, it was not possible to determine the effects of the litter treatments used in this experiment. However, these results do indicate that a down-time of 19 days between flocks should be sufficient to eliminate the risk of *C. jejuni* cross-contamination from re-using the litter. This is supported by the observation that even the untreated, inoculated litter did not appear to spread *C. jejuni*.

It is important to note that down-time may be significantly lower in a commercial setting and can be as little as 7 days between flocks [18]. In a past experiment, litter contaminated with *C. jejuni* by inoculated birds was shown to cross-contaminate a second flock raised on the same litter [10]. However, the down-time between flocks was only about 2 h in that case. In our experiment, a 19-day down-time was unavoidable due to Coronavirus-disease-2019-related supply issues. Because of this extended down-time, further experimentation is necessary to track *C. jejuni* populations in the litter throughout the down-time to pinpoint the minimum time necessary to eliminate *C. jejuni*. In addition, variables in the house and litter such as the temperature, moisture, and ammonia levels may influence the survivability of *C. jejuni* during down-time [19,20,21], therefore these should be investigated.

Another significant factor that may influence the detection of *C. jejuni* in litter is the formation of a viable but non-culturable (VBNC) state. It has been documented that *C. jejuni* cells may assume this state when subjected to stressful environmental conditions [22,23]. *C. jejuni* cells in the VBNC state would be difficult to detect, as they would not grow during enrichment [22,23,24]. It may be possible that some *C. jejuni* cells survived in the litter in the VBNC state and were not detected by our methods. However, because it appeared that the birds in the second flock were not colonized by *C. jejuni*, a potential VBNC state did not appear to be a significant factor influencing cross-contamination between flocks in this experiment, as previous experiments have demonstrated that VBNC *C. jejuni* isolates can be resuscitated by intestinal passage in a mouse model [24].

## 5. Conclusions

In this experiment, no determinations could be made on the effects of windrow composting for 19 days and sodium bisulfate litter treatments (45 kg/305 m^2^) on the cross-contamination of broilers by *C. jejuni* because no cross-contamination of the second flock was observed. Nonetheless, it was demonstrated that a sufficient down-time between flocks can prevent cross-contamination, which indicates that down-time is an important factor for mitigating *C. jejuni* in broilers. However, the necessary length of down-time and the effects of environmental conditions during this period remain unclear. It would be beneficial for future research to explore the potential effects of these variables on cross-contamination of *C. jejuni* between flocks.

## Figures and Tables

**Figure 1 animals-12-00858-f001:**
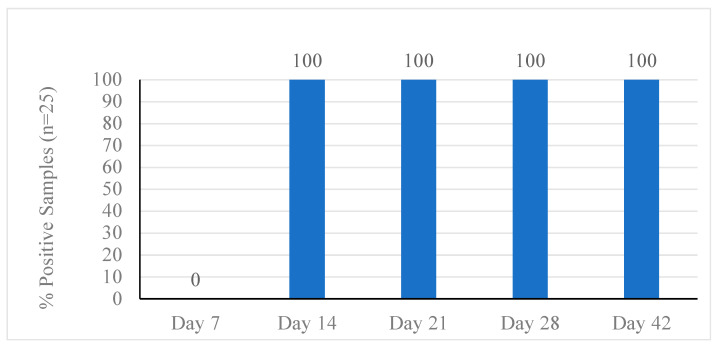
*Campylobacter jejuni* prevalence in boot cover swabs during grow-out of flock 1.

**Figure 2 animals-12-00858-f002:**
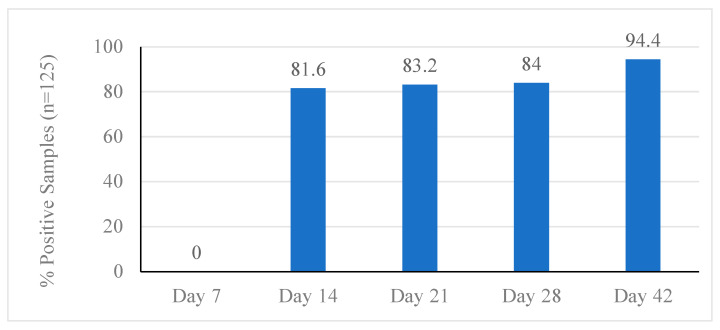
*Campylobacter jejuni* prevalence in ceca samples during grow-out of flock 1.

**Table 1 animals-12-00858-t001:** Nutrient composition of diets used for all birds in the study.

Diet	Crude Protein (%)	Metabolizable Energy (kcal/kg)
Starter	22.08	3053.82
Grower	20.01	3130.69
Finisher	17.63	3176.04

**Table 2 animals-12-00858-t002:** Litter treatments applied to used litter before placing second flock.

Treatment	Description	Birds Inoculated in First Flock
A	Negative control—fresh pine shavings	No
B	Positive control—used litter with no litter treatment	Yes
C	Used litter with sodium bisulfate applied prior to placing birds (45 kg/305 m^2^)	Yes
D	Used litter windrow composted between flocks 1 and 2	Yes
E	Used litter windrow composted between flocks and sodium bisulfate applied prior to placing birds (45 kg/305 m^2^)	Yes

## Data Availability

The data presented in this study are available on request from the corresponding author.

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
