# Peer review of "Effects of Common Litter Management Practices on the Prevalence of Campylobacter jejuni in Broilers"

_animals, 2022, doi:10.3390/ani12070858_

Round 1
Reviewer 1 Report
This is an interesting paper presenting relevant results.
Overall the paper is very clearly written ad the experimental design, as well as the results, are clearly presented
I only have a few minor issues:
it would be nice to describe shortly how windrow composting works as this is likely not known by everybody, especially not in those parts of the world where this methodology is not used.
in the description of the study design, it would be good to clarify how the pens were separated from each other? Given the fact that also the negative control pens became positive, it is expected that some cross-contamination between different pens/treatments occurred. It would be good to shortly address this in the discussion as well.
Author Response
Response to Comment 1:
A short description of windrow composting was added to the litter treatments section in the materials and methods section (section 2.4, lines 112-119).
Response to Comment 2:
A description of how the pens were separated from one another was added to the materials and methods section (section 2.1, lines 84-86). In addition, a statement addressing the cross-contamination of negative controls was added to the discussion section, as recommended.
Reviewer 2 Report
The manuscript „ Effects of common litter management practices on prevlence of Capylobacter jejuni in broilers“ investigated if the common treatments of contaminated litter with campylobacter could prevent cross-contamination. The authors concluded that the re-used of litter was not a significant reservoir for cross-contamination when provided a significant down-time (19 days in this study) between flocks. The manuscript is very interesting and well written.
Moderate Revision:
- The authors did not pinpoint the minimum time to eliminate Capylobacter jejuni, discussed as a limitation of the study
- The type of bedding materials and other factors such as pH, composition, water holding capacity, buffer capacity, depth, particle size, moisture content and microbial community might be also considered. Please discuss and cite this manuscript: doi.org/10.51585/gjvr.2021.3.0017 and this MS doi.vorg/10.1007/s11356-021-17613-0
- The novelty of this work should be focused on the introduction section.
- In the abstract and conclusion section, the dose of sodium bisulfate and down time between foclks should be given.
- Line 79: please add the full name of CFU (Colony forming unit)
- Line 178: Covid-19 should be Coronvirus diseases 2019 (COVID-19)
- There is no statistical analysis to support the significance of this work, and it should be?
- The form of feeds and feed composition and nutrients profile table should be added.
- The husbandry and Vet care details and any antibiotics or AGPs used should be declared in this kind of the MS.
Author Response
Response to Comment 1
We acknowledge that not pinpointing a more precise time to eliminate C. jejuni is indeed a limitation of this study. However, a composting time of 19 days to completely eliminate C. jejuni from litter is still useful to publish for future researchers as a starting point.
Response to Comment 2
The possible effects of litter/house variables such as moisture, pH, and temperature on C. jejuni survivability in litter are mentioned in the discussion section (lines 201-203).
Response to Comment 3
A statement was added to lines 69-72 of the introduction to highlight the novelty of this experiment.
Response to Comment 4
Done.
Response to Comment 5
Done.
Response to Comment 6
Done.
Response to Comment 7
No statistical analysis was possible because no C. jejuni was observed in the second flock. There was no data to compare.
Response to Comment 8
Form of feeds (crumbles/pellets) was added to the materials and methods section (section 2.1, lines 80-81). Table 1 was added to list CP and ME for all diets.
Response to Comment 9
This experiment was conducted in accordance with the Auburn University Institutional Animal Care and Use Committee, and a statement has been provided in the manuscript. The IACUC committee requires proper husbandry and veterinary care for all experimental animals under its jurisdiction. No antibiotics or growth-promoting feed additives were used in this experiment.
Round 2
Reviewer 2 Report
The authors have made good progress revising the MS, however, some points must be declared before acceptance:
- There are no statistical analyses for the data collected and this must be declared.
- The condition and characteristics of crumbles and pelleted have to be added.
- L 172-174, authors stated that the fact that the negative control pens were positive although they were not inoculated with C. jejuni also illustrates the rapid spread of this organism and shows that separation by pens within a house is not sufficient to prevent cross-contamination. I suggest that the authors should demonstrate microbial swabs of pens prior to the introduction of the birds, or else the hygiene/ sanitation process of the pens prior to housing may be insufficient, and this should be pointed out.
- References can be updated with Literature published so far in 2022 and with the following: Attia Youssef A., Fulvia Bovera, · Reda A. Hassan, · Ebtehal A. Hassan, · Khalil M. Attia, · Mohamed H. Assar and · Fouad A. Tawfeek (2021). Reducing ammonia emission by aluminum sulfate addition in litter and its influence on productive, reproductive, and physiological parameters of dual‑purpose breeding hens. Environmental Science and Pollution Research https://doi.org/10.1007/s11356-021-17613-0
- Thank you and best wishes